# Linguistically Regularized LSTM for Sentiment Classification

## Abstract

This paper deals with sentence-level sentiment classification. Though a variety of neural network models have been proposed recently, however, previous models either depend on expensive phrase-level annotation, most of which has remarkably degraded performance when trained with only sentence-level annotation; or do not fully employ linguistic resources (e.g., sentiment lexicons, negation words, intensity words). In this paper, we propose simple models trained with sentence-level annotation, but also attempt to model the linguistic role of sentiment lexicons, negation words, and intensity words. Results show that our models are able to capture the linguistic role of sentiment words, negation words, and intensity words in sentiment expression.

## 1 Introduction

Sentiment classification aims to classify text to sentiment classes such as *positive or negative*, or more fine-grained classes such as *very positive, positive, neutral, etc*. There has been a variety of approaches for this purpose such as lexicon-based classification (Turney, 2002; Taboada et al., 2011), and early machine learning based methods (Pang et al., 2002; Pang and Lee, 2005), and recently neural network models such as convolutional neural network (CNN) (Kim, 2014; Kalchbrenner et al., 2014; Lei et al., 2015), recursive autoencoders (Socher et al., 2011, 2013), Long Short-Term Memory (LSTM) (Mikolov, 2012; Chung et al., 2014; Tai et al., 2015; Zhu et al., 2015), and many more.

In spite of the great success of these neural models, there are some defects in previous studies.

First, tree-structured models such as recursive autoencoders and Tree-LSTM (Tai et al., 2015; Zhu et al., 2015), depend on parsing tree structures and expensive phrase-level annotation, whose performance drops substantially when only trained with sentence-level annotation. Second, linguistic knowledge such as sentiment lexicon, negation words or *negators* (e.g., *not, never*), and intensity words or *intensifiers* (e.g., *very, absolutely*), has not been fully employed in neural models.

The goal of this research is to developing simple sequence models but also attempts to fully employing linguistic resources to benefit sentiment classification. Firstly, we attempts to develop simple models that do not depend on parsing trees and do not require phrase-level annotation which is too expensive in real-world applications. Secondly, in order to obtain competitive performance, simple models can benefit from linguistic resources. Three types of resources will be addressed in this paper: sentiment lexicon, negation words, and intensity words. Sentiment lexicon offers the prior polarity of a word which can be useful in determining the sentiment polarity of longer texts such as phrases and sentences. Negators are typical sentiment shifters (Zhu et al., 2014), which constantly change the polarity of sentiment expression. Intensifiers change the valence degree of the modified text, which is important for fine-grained sentiment classification.

In order to model the linguistic role of sentiment, negation, and intensity words, our central idea is to regularize the difference between the predicted sentiment distribution of the current position [1], and that of the previous or next positions, in a sequence model. For instance, if the current position is a negator *not*, the negator should change the sentiment distribution of the next posi-

---

[1]Note that in sequence models, the hidden state of the current position also encodes forward or backward contexts.

tion accordingly. To summarize, our contributions lie in two folds:

- We discover that modeling the linguistic role of sentiment, negation, and intensity words can enhance sentence-level sentiment classification. We address the issue by imposing linguistic-inspired regularizers on sequence LSTM models.

- Unlike previous models that depend on parsing structures and expensive phrase-level annotation, our models are simple and efficient, but the performance is on a par with the state-of-the-art.

The rest of the paper is organized as follows: In the following section, we survey related work. In Section 3, we briefly introduce the background of LSTM and bidirectional LSTM, and then describe in detail the lingistic regularizers for sentiment/negation/intensity words in Section 4. Experiments are presented in Section 5, and Conclusion follows in Section 6.

## 2 Related Work

### 2.1 Neural Networks for Sentiment Classification

There are many neural networks proposed for sentiment classification. The most noticeable models may be the recursive autoencoder neural network which builds the representation of a sentence from subphrases recursively (Socher et al., 2011, 2013; Dong et al., 2014; Qian et al., 2015). Such recursive models usually depend on a tree structure of input text, and in order to obtain competitive results, usually require annotation of all subphrases. Sequence models, for instance, convolutional neural network (CNN), do not require tree-structured data, which are widely adopted for sentiment classification (Kim, 2014; Kalchbrenner et al., 2014; Lei et al., 2015). Long short-term memory models are also common for learning sentence-level representation due to its capability of modeling the prefix or suffix context (Hochreiter and Schmidhuber, 1997). LSTM can be commonly applied to sequential data but also tree-structured data (Zhu et al., 2015; Tai et al., 2015).

### 2.2 Applying Linguistic Knowledge for Sentiment Classification

Linguistic knowledge and sentiment resources, such as sentiment lexicons, negation words (*not,*

*never, neither, etc.*) or *negators*, and intensity words (*very, extremely, etc.*) or *intensifiers*, are useful for sentiment analysis in general.

Sentiment lexicon (Hu and Liu, 2004; Wilson et al., 2005) usually defines prior polarity of a lexical entry, and is valuable for lexicon-based models (Turney, 2002; Taboada et al., 2011), and machine learning approaches (Pang and Lee, 2008). There are recent works for automatic construction of sentiment lexicons from social data (Vo and Zhang, 2016) and for multiple languages (Chen and Skiena, 2014). A noticeable work that ultilizes sentiment lexicons can be seen in (Teng et al., 2016) which treats the sentiment score of a sentence as a weighted sum of prior sentiment scores of negation words and sentiment words, where the weights are learned by a neural network.

Negation words play a critical role in modifying sentiment of textual expressions. Some early negation models adopt the *reversing assumption* that a negator reverses the sign of the sentiment value of the modified text (Polanyi and Zaenen, 2006; Kennedy and Inkpen, 2006). The *shifting hyothesis* assumes that negators change the sentiment values by a constant amount (Taboada et al., 2011; Liu and Seneff, 2009). Since each negator can affect the modified text in different ways, the constant amount can be extended to be negator-specific (Zhu et al., 2014), and further, the effect of negators could also depend on the syntax and semantics of the modified text (Zhu et al., 2014). Other approaches to negation modeling can be seen in (Jia et al., 2009; Wiegand et al., 2010; Benamara et al., 2012; Lapponi et al., 2012).

Sentiment intensity of a phrase indicates the strength of associated sentiment, which is quite important for fine-grained sentiment classification or rating. Intensity words can change the valence degree (i.e., sentiment intensity) of the modified text. In (Wei et al., 2011) the authors propose a linear regression model to predict the valence value for content words. In (Malandrakis et al., 2013), a kernel-based model is proposed to combine semantic information for predicting sentiment score. In the SemEval-2016 task 7 subtask A, a learning-to-rank model with a pair-wise strategy is proposed to predict sentiment intensity scores (Wang et al., 2016). Linguistic intensity is not limited to sentiment or intensity words, and there are works that assign low/medium/high intensity scales to adjectives such as *okay, good, great* (Sharma et al.,

2015) or to gradable terms (e.g. *large, huge, gigantic*) (Shivade et al., 2015).

In (Dong et al., 2015), a sentiment parser is proposed, and the authors studied how sentiment changes when a phrase is modified by negators or intensifiers.

## 3 Long Short-term Memory Network

### 3.1 Long Short-Term Memory (LSTM)

Long Short-Term Memory has been widely adopted for text processing. Briefly speaking, in LSTM, the hidden states $h_t$ and memory cell $c_t$ is a function of their previous $c_{t-1}$ and $h_{t-1}$ and input vector $x_t$, or formally as follows:

$$c_t, h_t = g^{(LSTM)}(c_{t-1}, h_{t-1}, x_t) \qquad (1)$$

The hidden state $h_t \in R^d$ denotes the representation of position $t$ while also encoding the preceding contexts of the position. For more details about LSTM, we refer readers to (Hochreiter and Schmidhuber, 1997).

### 3.2 Bidirectional LSTM

In LSTM, the hidden state of each position $(h_t)$ only encodes the prefix context in a forward direction while the backward context is not considered. Bidirectional LSTM (Graves et al., 2013) exploited two parallel passes (forward and backward) and concatenated hidden states of the two LSTMs as the representation of each position. The forward and backward LSTMs are respectively formulated as follows:

$$\overrightarrow{c}_t, \overrightarrow{h}_t = g^{(LSTM)}(\overrightarrow{c}_{t-1}, \overrightarrow{h}_{t-1}, x_t) \qquad (2)$$

$$\overleftarrow{c}_t, \overleftarrow{h}_t = g^{(LSTM)}(\overleftarrow{c}_{t+1}, \overleftarrow{h}_{t+1}, x_t) \qquad (3)$$

where $g^{(LSTM)}$ is the same as that in Eq (1). Particularly, parameters in the two LSTMs are shared. The representation of the entire sentence is $[\overrightarrow{h}_n, \overleftarrow{h}_1]$, where $n$ is the length of the sentence. At each position $t$, the new representation is $h_t = [\overrightarrow{h}_t, \overleftarrow{h}_t]$, which is the concatenation of hidden states of the forward LSTM and backward LSTM. In this way, the forward and backward contexts can be considered simultaneously.

## 4 Linguistically Regularized LSTM

The central idea of the paper is to model the linguistic role of sentiment, negation, and intensity words in sentence-level sentiment classification by regularizing the outputs at adjacent positions of a sentence. For example, in sentence "*this movie is interesting*", the predicted sentiment distributions at "*this\**[2]", "*this movie\**", and "*this movie is\**" should be close to each other, while the predicted sentiment distribution at "*this movie is very interesting\**" should be quite different from the preceeding positions ("*this movie is very\**") since a sentiment word ("*interesting*") is seen.

We propose a generic regularizer and three special regularizers based on the following linguistic observations:

- **Non-Sentiment Regularizer**: if the two adjacent positions are all non-opinion words, the sentiment distributions of the two positions should be close to each other. Though this is not always true (e.g., *soap movie*), this assumption holds at most cases.

- **Sentiment Regularizer**: if the word is a sentiment word found in a lexicon, the sentiment distribution of the current position should be significantly different from that of the next or previous positions. We approach this phenomenon with a *sentiment class specific shifting distribution*.

- **Negation Regularizer**: Negation words such as "not" and "never" are critical sentiment shifter or converter: in general they shift sentiment polarity from the positive end to the negative end, but sometimes depend on the negation word and the words they modify. The negation regularizer models this linguistic phenomena with a *negator-specific transformation matrix*.

- **Intensity Regularizer**: Intensity words such as "very" and "extremely" change the valence degree of a sentiment expression: for instance, from *positive* to *very positive*. Modeling this effect is quite important for fine-grained sentiment classification, and the intensity regularizer is designed to formulate this effect by a *word-specific transformation matrix*.

More formally, the predicted sentiment distribution ($p_t$, based on $h_t$, see Eq. 5) at position $t$ should be linguistically regularized with respect to that of the preceding ($t - 1$) or following ($t + 1$)

---

[2]The asterisk denotes the current position.

positions. In order to enforce the model to produce coherent predictions, we plug a new loss term into the original cross entropy loss:

$$\mathcal{L}(\theta) = -\sum_i y^i \log p^i + \alpha \sum_i \sum_t L_{t,i} + \beta||\theta||^2 \quad (4)$$

where $y^i$ is the gold distribution for sentence $i$, $p^i$ is the predicted distribution, $L_{t,i}$ is one of the above regularizers or combination of these regularizers on sentence $i$, $\alpha$ is the weight for the regularization term, and $t$ is the word position in a sentence.

**Note that** we do not consider the modification span of negation and intensity words to preserve the simplicity of the proposed models. Negation scope resolution is another complex problem which has been extensively studied (Zou et al., 2013; Packard et al., 2014; Fancellu et al., 2016), which is beyond the scope of this work. Instead, we resort to sequence LSTMs for encoding surrounding contexts at a given position.

### 4.1 Non-Sentiment Regularizer (NSR)

This regularizer constrains that the sentiment distributions of adjacent positions should not vary much if the additional input word $x_t$ is not a sentiment word, formally as follows:

$$L_t^{(NSR)} = max(0, D_{KL}(p_t||p_{t-1}) - M) \quad (5)$$

where $M$ is a hyperparameter for margin, $p_t$ is the predicted distribution at state of position $t$, (i.e., $h_t$), and $D_{KL}(p||q)$ is a symmetric KL divergence defined as follows:

$$D_{KL}(p||q) = \frac{1}{2} \sum_{l=1}^{C} p(l) \log q(l) + q(l) \log p(l) \quad (6)$$

where $p, q$ are distributions over sentiment labels $l$ and $C$ is the number of labels.

### 4.2 Sentiment Regularizer (SR)

The sentiment regularizer constrains that the sentiment distributions of adjacent positions should drift accordingly if the input word is a sentiment word. Let's revisit the example "*this movie is interesting*" again. At position $t = 4$ we see a positive word "*interesting*" so the predicted distribution would be more positive than that at position $t = 3$. This is the issue of *sentiment drift*.

In order to address the sentiment drift issue, we propose a polarity shifting distribution $s_c \in R^C$ for each sentiment class defined in a lexicon. For

instance, a sentiment lexicon may have class labels like *strong positive*, *weakly positive*, *weakly negative*, and *strong negative*, and for each class, there is a shifting distribution which will be learned by the model. The sentiment regularizer states that if the current word is a sentiment word, the sentiment distribution drift should be observed in comparison to the previous position, in more details:

$$p_{t-1}^{(SR)} = p_{t-1} + s_{c(x_t)} \quad (7)$$

$$L_t^{(SR)} = max(0, D_{KL}(p_t||p_{t-1}^{(SR)}) - M) \quad (8)$$

where $p_{t-1}^{(SR)}$ is the drifted sentiment distribution after considering the shifting sentiment distribution corresponding to the state at position $t$, $c(x_t)$ is the prior sentiment class of word $x_t$, and $s_c \in \theta$ is a parameter to be optimized but could also be set fixed with prior knowledge. Note that in this way all words of the same sentiment class share the same drifting distribution, but in a refined setting, we can learn a shifting distribution for each sentiment word if large-scale datasets are available.

### 4.3 Negation Regularizer (NR)

The negation regularizer approaches how negation words shift the sentiment distribution of the modified text. When the input $x_t$ is a negation word, the sentiment distribution should be shifted/reversed accordingly. However, the negation role is more complex than that by sentiment words, for example, the word "*not*" in "*not good*" and "*not bad*" have different roles in polarity change. The former changes the polarity to *negative*, while the latter changes to *neutral* instead of *positive*.

To respect such complex negation effects, we propose a transformation matrix $T_m \in R^{C \times C}$ for each negation word $m$, and the matrix will be learned by the model. The regularizer assumes that if the current position is a negation word, the sentiment distribution of the current position should be close to that of the next or previous position with the transformation.

$$p_{t-1}^{(NR)} = softmax(T_{x_j} \times p_{t-1}) \quad (9)$$

$$p_{t+1}^{(NR)} = softmax(T_{x_j} \times p_{t+1}) \quad (10)$$

$$L_t^{(NR)} = min \begin{cases} max(0, D_{KL}(p_t||p_{t-1}^{(NR)}) - M) \\ max(0, D_{KL}(p_t||p_{t+1}^{(NR)}) - M) \end{cases} \quad (11)$$

where $p_{t-1}^{(NR)}$ and $p_{t+1}^{(NR)}$ is the sentiment distuibution after transformation, $T_{x_j} \in \theta$ is the transformation matrix for a negation word $x_j$, a parameter to be learned during training. In total, we train $m$ transformation matrixs for $m$ negation words. Such negator-specific transformation is in accordance with the finding that each negator has its individual negation effect (Zhu et al., 2014).

## 4.4 Intensity Regularizer (IR)

Sentiment intensity of a phrase indicates the strength of associated sentiment, which is quite important for fine-grained sentiment classification or rating. Intensifier can change the valence degree of the content word. The intensity regularizer models how intensity words influence the sentiment valence of a phrase or a sentence.

The formulation of the intensity effect is quite the same as that in the negation regularizer, but with different parameters of course. For each intensity word, there is a transform matrix to favor the different roles of various intensifiers on sentiment drift. For brevity, we will not repeat the formulas here.

## 4.5 Applying Linguistic Regularizers to Bidirectional LSTM

To preserve the simplicity of our proposals, we do not consider the modification span of negation and intensity words, which is a quite challenging problem in the NLP community (Zou et al., 2013; Packard et al., 2014; Fancellu et al., 2016). However, we can alleviate the problem by leveraging bidirectional LSTM.

For a single LSTM, we employ a backward LSTM from the end to the beginning of a sentence. This is because, at most times, the modified words of negation and intensity words are usually at the right side of the modified text. But sometimes, the modified words are at the left side of negation and intensity words. To better address this issue, we employ bidirectional LSTM and let the model determine which side should be chosen.

More formally, in Bi-LSTM, we compute a transformed sentiment distribution on $\overrightarrow{p}_{t-1}$ of the forward LSTM and also that on $\overleftarrow{p}_{t+1}$ of the backward LSTM, and compute the minimum distance of the distribution of the current position to the two distributions. This could be formulated as follows:

$$\overrightarrow{p}_{t-1}^{(R)} = softmax(T_{x_j} \times \overrightarrow{p}_{t-1}) \quad (12)$$

$$\overleftarrow{p}_{t+1}^{(R)} = softmax(T_{x_j} \times \overleftarrow{p}_{t+1}) \quad (13)$$

$$L_t^{(R)} = min \begin{cases} max(0, D_{KL}(\overrightarrow{p}_t || \overrightarrow{p}_{t-1}^{(R)}) - M) \\ max(0, D_{KL}(\overleftarrow{p}_t || \overleftarrow{p}_{t+1}^{(R)}) - M) \end{cases} \quad (14)$$

where $\overrightarrow{p}_{t-1}^{(R)}$ and $\overleftarrow{p}_{t+1}^{(R)}$ are the sentiment distributions transformed from the previous distribution $\overrightarrow{p}_{t-1}$ and next distribution $\overleftarrow{p}_{t+1}$ respectively. Note that $R \in \{NR, IR\}$ indicating the formulation works for both negation and intensity regularizers.

Due to the same consideration, we redefine $L_t^{(NSR)}$ and $L_t^{(SR)}$ with bidirectional LSTM similarly. The formulation is the same and omitted for brevity.

## 4.6 Discussion

Our models address these linguistic factors with mathematical operations, parameterized with shifting distribution vectors or transformation matrices. In the sentiment regularizer, the sentiment shifting effect is parameterized with a *class-specific* distribution (but could also be word-specific if with more data). In the negation and intensity regularizers, the effect is parameterized with *word-specific* transformation matrices. This is to respect the fact that the mechanism of how negation and intensity words shift sentiment expression is quite complex and highly dependent on individual words. Negation/Intensity effect also depends on the syntax and semantics of the modified text, however, for simplicity we resort to sequence LSTM for encoding surrounding contexts in this paper. We partially address the modification scope issue by applying the minimization operator in Eq. 11 and Eq. 14, and the bidirectional LSTM.

## 5 Experiment

### 5.1 Dataset and Sentiment Lexicon

Two datasets are used for evaluating the proposed models: Movie Review (MR) (Pang and Lee, 2005) where each sentence is annotated with two classes as *negative, positive* and Stanford Sentiment Treebank (SST) (Socher et al., 2013) with five classes { *very negative, negative, neutral, positive, very positive*}. Note that SST has provided phrase-level annotation on all inner nodes, but we only use the sentence-level annotation since one of our goals is to avoid expensive phrase-level annotation.

The sentiment lexicon contains two parts. The first part comes from MPQA (Wilson et al., 2005), which contains $5,153$ sentiment words, each with polarity rating. The second part consists of the leaf nodes of the SST dataset (i.e., all sentiment words) and there are $6,886$ polar words except *neural* ones. We combine the two parts and ignore those words that have conflicting sentiment labels, and produce a lexicon of $9,750$ words with 4 sentiment labels. For negation and intensity words, we collect them manually since the number is small, some of which can be seen in Table 2.

Due to the length limit, we present the implementation details and a full list of resources in the supplementary file.

| Dataset | MR | SST |
|---|---|---|
| # sentences in total | 10,662 | 11,885 |
| #sen containing sentiment word | 10,446 | 11,211 |
| #sen containing negation word | 1,644 | 1,832 |
| #sen containing intensity word | 2,687 | 2,472 |

Table 1: The data statistics.

| Negation word | no, nothing, never, neither, not, seldom, scarcely, etc. |
|---|---|
| Intensity word | terribly, greatly, absolutely, too, very, completely, etc. |

Table 2: Examples of negation and intensity words.

### 5.2 Overall Comparison

We include several baselines, as listed below:

**RNN/RNTN**: Recursive Neural Network over parsing trees, proposed by (Socher et al., 2011) and Recursive Tensor Neural Network (Socher et al., 2013) employs tensors to model correlations between different dimensions of child nodes' vectors.

**LSTM/Bi-LSTM**: Long Short-Term Memory (Cho et al., 2014) and the bidirectional variant as introduced previously.

**Tree-LSTM**: Tree-Structured Long Short-Term Memory (Tai et al., 2015) introduces memory cells and gates into tree-structured neural network.

**CNN**: Convolutional Neural Network (Kalchbrenner et al., 2014) generates sentence representation by convolution and pooling operations.

**CNN-Tensor**: In (Lei et al., 2015), the convolution operation is replaced by tensor product and

a dynamic programming is applied to enumerate all skippable trigrams in a sentence. Very strong results are reported.

**DAN**: Deep Average Network (DAN) (Iyyer et al., 2015) averages all word vectors in a sentence and connects an MLP layer to the output layer.

**Neural Context-Sensitive Lexicon: NCSL** (Teng et al., 2016) treats the sentiment score of a sentence as a weighted sum of prior scores of words in the sentence where the weights are learned by a neural network.

Firstly, we evaluate our model on the MR dataset and the results are shown in Table 3. We have the following observations:

**First**, both LR-LSTM and LR-Bi-LSTM outperforms their counterparts (81.5% vs. 77.4% and 82.1% vs. 79.3%, resp.), demonstrating the effectiveness of the linguistic regularizers. **Second**, LR-LSTM and LR-Bi-LSTM perform slightly better than Tree-LSTM but Tree-LSTM leverages a constituency tree structure while our model is a simple sequence model. As future work, we will apply such regularizers to tree-structured models. **Last**, on the MR dataset, our model is comparable to or slightly better than CNN.

For fine-grained sentiment classification, we evaluate our model on the SST dataset which has five sentiment classes { *very negative, negative, neutral, positive, very positive*} so that we can evaluate the sentiment shifting effect of intensity words. The results are shown in Table 3. We have the following observations:

**First**, linguistically regularized LSTM and Bi-LSTM are better than their counterparts. It's worth noting that LR-Bi-LSTM (trained with just sentence-level annotation) is even comparable to Bi-LSTM trained with phrase-level annotation. That means, LR-Bi-LSTM can avoid the heavy phrase-level annotation but still obtain comparable results. **Second**, our models are comparable to Tree-LSTM but our models are not dependent on a parsing tree and more simple, and hence more efficient. Further, for Tree-LSTM, the model is heavily dependent on phrase-level annotation, otherwise the performance drops substantially (from 51% to 48.1%). **Last**, on the SST dataset, our model is better than CNN, DAN, and NCSL. We conjecture that the strong performance of CNN-Tensor may be due to the tensor product operation, the enumeration of all skippable trigrams, and the

concatenated representations of all pooling layers for final classification.

| Method | MR | SST phrase-level | SST sentence-level |
|---|---|---|---|
| RNN | 77.7* | 44.8# | 43.2* |
| RNTN | 75.9# | 45.7* | 43.4# |
| LSTM | 77.4# | 46.4* | 45.6# |
| Bi-LSTM | 79.3# | 49.1* | 46.5# |
| Tree-LSTM | 80.7# | 51.0* | 48.1# |
| CNN | 81.5* | 48.0* | 46.9# |
| CNN-Tensor | - | 51.2* | 50.6* |
| DAN | - | - | 47.7* |
| NCSL | - | 51.1* | 47.1# |
| LR-Bi-LSTM | 82.1 | - | 48.6 |
| LR-LSTM | 81.5 | - | 48.2 |

Table 3: The accuracy on MR and SST. *Phrase-level* means the models use phrase-level annotation for training. And *sentence-level* means the models only use sentence-level annotation. Results marked with * are re-printed from the references, while those with # are obtained either by our own implementation or with the same codes shared by the original authors.

## 5.3 The Effect of Different Regularizers

In order to reveal the effect of each individual regularizer, we conduct ablation experiments. Each time, we remove a regularizer and observe how the performance varies. First of all, we conduct this experiment on the entire datasets, and then we experiment on sub-datasets that only contain negation words or intensity words.

The experiment results are shown in Table 4 where we can see that the non-sentiment regularizer (NSR) and sentiment regularizer (SR) play a key role[3], and the negation regularizer and intensity regularizer are effective but less important than NSR and SR. This may be due to the fact that only 14% of sentences contains negation words in the test datasets, and 23% contains intensity words, and thus we further evaluate the models on two subsets, as shown in Table 5.

The experiments on the subsets show that: 1) With linguistic regularizers, LR-Bi-LSTM outperforms Bi-LSTM remarkably on these subsets; 2) When the negation regularizer is removed from the model, the performance drops significantly on

---

[3]Kindly note that almost all sentences contain sentiment words, see Tab. 1.

| Method | MR | SST |
|---|---|---|
| LR-Bi-LSTM | 82.1 | 48.6 |
| LR-Bi-LSTM (-NSR) | 80.8 | 46.9 |
| LR-Bi-LSTM (-SR) | 80.6 | 46.9 |
| LR-Bi-LSTM (-NR) | 81.2 | 47.6 |
| LR-Bi-LSTM (-IR) | 81.7 | 47.9 |
| LR-LSTM | 81.5 | 48.2 |
| LR-LSTM (-NSR) | 80.2 | 46.4 |
| LR-LSTM (-SR) | 80.2 | 46.6 |
| LR-LSTM (-NR) | 80.8 | 47.4 |
| LR-LSTM (-IR) | 81.2 | 47.4 |

Table 4: The accuracy for LR-Bi-LSTM and LR-LSTM with regularizer ablation. *NSR*, *SR*, *NR* and *IR* denotes *Non-sentiment Regularizer*, *Sentiment Regularizer*, *Negation Regularizer*, and *Intensity Regularizer* respectively.

both MR and SST subsets; 3) Similar observations can be found regarding the intensity regularizer.

| Method | Neg. Sub. MR | Neg. Sub. SST | Int. Sub. MR | Int. Sub. SST |
|---|---|---|---|---|
| BiLSTM | 72.0 | 39.8 | 83.2 | 48.8 |
| LR-Bi-LSTM (-NR) | 74.2 | 41.6 | - | - |
| LR-Bi-LSTM (-IR) | - | - | 85.2 | 50.0 |
| LR-Bi-LSTM | 78.5 | 44.4 | 87.1 | 53.2 |

Table 5: The accuracy on the negation sub-dataset (Neg. Sub.) that only contains negators, and intensity sub-dataset (Int. Sub.) that only contains intensifiers.

## 5.4 The Effect of the Negation Regularizer

To further reveal the linguistic role of negation words, we compare the predicted sentiment distributions of a phrase pair with and without a negation word. The experimental results performed on MR are shown in Fig. 1. Each dot denotes a phrase pair (for example, $<interesting, not\ interesting>$), where the x-axis denotes the positive score[4] of a phrase without negators (e.g., *interesting*), and the y-axis indicates the positive score for the phrase with negators (e.g., *not interesting*). The curves in the figures show this function: $[1 - y, y] = softmax(T_{nw} * [1 - x, x])$ where $[1 - x, x]$ is a sentiment distribution on $[negative, positive]$, $x$ is the positive score of the phrase without negators (x-axis) and $y$ that of the phrase with negators (y-

---

[4] The score is obtained from the predicted distribution, where 1 means positive and 0 means negative.

axis), and $T_{nw}$ is the transformation matrix for the negation word $nw$ (see Eq. 9). By looking into the

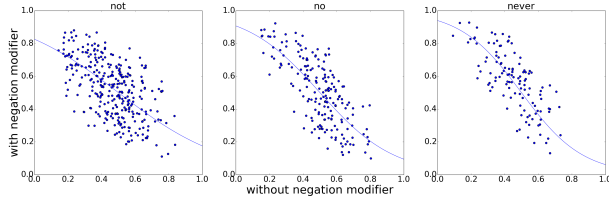

Figure 1: The sentiment shifts with negators. Each dot $< x, y >$ indicates that $x$ is the sentiment score of a phrase without negator and $y$ is that of the phrase with a negator.

detailed results of our model, we have the following statements:

**First**, there is no dot at the up-right and bottom-left blocks, indicating that negators generally shift/convert very positive or very negative phrases to other polarities. Typical phrases include *not very good, not too bad*.

**Second,** the dots at the up-left and bottom-right respectively indicates the negation effects: changing negative to positive and positive to negative. Typical phrases include *never seems hopelessly (up-left), no good scenes (bottom-right), not interesting (bottom-right)*, etc. There are also some positive/negative phrases shifting to neutral sentiment such as *not so good*, and *not too bad*.

**Last,** the dots located at the center indicate that neutral phrases maintain neutral sentiment with negators. Typical phrases include *not at home, not here*, where negators typically modify non-sentiment words.

### 5.5 The Effect of the Intensity Regularizer

To further reveal the linguistic role of intensity words, we perform experiments on the SST dataset, as illustrated in Figure 2. We show the matrix that indicates how the sentiment shifts after being modified by intensifiers. Each number in a cell ($m_{ij}$) indicates how many phrases are predicted with a sentiment label $i$ but the prediction of the phrases with intensifiers changes to label $j$. For instance, the number 20 ($m_{21}$) in the second matrix , means that there are 20 phrases predicted with a class of *negative* (-) but the prediction changes to *very negative* (- -) after being modified by intensifier *"very"*. Results in the first matrix show that, for intensifier *"most"*, there are 21/21/13/12 phrases whose sentiment is shifted after being modified by intensifiers, from negative

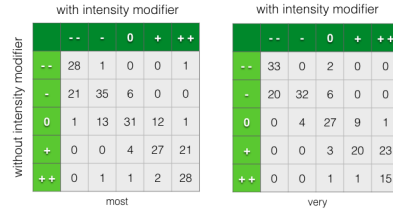

| with intensity modifier (most) | -- | - | 0 | + | ++ |
|---|---|---|---|---|---|
| -- | 28 | 1 | 0 | 0 | 1 |
| - | 21 | 35 | 6 | 0 | 0 |
| 0 | 1 | 13 | 31 | 12 | 1 |
| + | 0 | 0 | 4 | 27 | 21 |
| ++ | 0 | 1 | 1 | 2 | 28 |

| with intensity modifier (very) | -- | - | 0 | + | ++ |
|---|---|---|---|---|---|
| -- | 33 | 0 | 2 | 0 | 0 |
| - | 20 | 32 | 6 | 0 | 0 |
| 0 | 0 | 4 | 27 | 9 | 1 |
| + | 0 | 0 | 3 | 20 | 23 |
| ++ | 0 | 0 | 1 | 1 | 15 |

Figure 2: The sentiment shifting with intensifiers. The number in cell($m_{ij}$) indicates how many phrases are predicted with sentiment label $i$ but the prediction of phrases with intensifiers changes to label $j$.

to very negative (*eg. most irresponsible picture*), positive to very positive (*eg. most famous author*), neutral to negative (*eg. most plain*), and neutral to positive (*eg. most closely*), respectively.

There are also many phrases retaining the sentiment after being modified with intensifiers. Not surprisingly, for very positive/negative phrases, phrases modified by intensifiers still maintain the strong sentiment. For the left phrases, they fall into three categories: first, words modified by intensifiers are non-sentiment words, such as *most of us, most part*; second, intensifiers are not strong enough to shift sentiment, such as *most complex* (from neg. to neg.), *most traditional* (from *pos.* to *pos.*); third, our models fail to shift sentiment with intensifiers such as *most vital, most resonant film*.

## 6 Conclusion and Future Work

We present linguistically regularized LSTMs for sentence-level sentiment classification. The proposed models address the sentient shifting effect of sentiment, negation, and intensity words. Furthermore, our models are sequence LSTMs which do not depend on a parsing tree-structure and do not require expensive phrase-level annotation. Results show that our models are able to address the linguistic role of sentiment, negation, and intensity words.

To preserve the simplicity of the proposed models, we do not consider the modification scope of negation and intensity words, though we partially address this issue by applying a minimization operartor (see Eq. 11, Eq. 14) and bi-directional LSTM. As future work, we plan to apply the linguistic regularizers to tree-LSTM to address the scope issue since the parsing tree is easier to indicate the modification scope explicitly.

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
