# Peer review of "Linguistically Regularized LSTM for Sentiment Classification"

_ACL 2017 — decision unknown_

[Official Review · Reviewer 1 · rating 3 · confidence 4]
soundness 5 · originality 3 · clarity 3 · impact 3 · substance 3 · appropriateness 5 · meaningful comparison 5 · presentation format Poster

Strengths:

- Innovative idea: sentiment through regularization
- Experiments appear to be done well from a technical point of view
- Useful in-depth analysis of the model

Weaknesses:

- Very close to distant supervision
- Mostly poorly informed baselines

General Discussion:

This paper presents an extension of the vanilla LSTM model that
incorporates sentiment information through regularization.  The
introduction presents the key claims of the paper: Previous CNN
approaches are bad when no phrase-level supervision is present.
Phrase-level annotation is expensive. The contribution of this paper is
instead a "simple model" using other linguistic resources.

The related work section provides a good review of sentiment
literature. However, there is no mention of previous attempts at
linguistic regularization (e.g., [YOG14]).

The explanation of the regularizers in section 4 is rather lengthy and
repetitive. The listing on p. 3 could very well be merged with the
respective subsection 4.1-4.4. Notation in this section is inconsistent
and generally hard to follow. Most notably, p is sometimes used with a
subscript and sometimes with a superscript.  The parameter \beta is
never explicitly mentioned in the text. It is not entirely clear to me
what constitutes a "position" t in the terminology of the paper. t is a
parameter to the LSTM output, so it seems to be the index of a
sentence. Thus, t-1 is the preceding sentence, and p_t is the prediction
for this sentence. However, the description of the regularizers talks
about preceding words, not sentences, but still uses. My assumption here
is that p_t is actually overloaded and may either mean the sentiment of
a sentence or a word. However, this should be made clearer in the text.

One dangerous issue in this paper is that the authors tread a fine line
between regularization and distant supervision in their work. The
problem here is that there are many other ways to integrate lexical
information from about polarity, negation information, etc. into a model
(e.g., by putting the information into the features). The authors
compare against a re-run or re-implementation of Teng et al.'s NSCL
model. Here, it would be important to know whether the authors used the
same lexicons as in their own work. If this is not the case, the
comparison is not fair. Also, I do not understand why the authors cannot
run NSCL on the MR dataset when they have access to an implementation of
the model. Would this not just be a matter of swapping the datasets? The
remaining baselines do not appear to be using lexical information, which
makes them rather poor. I would very much like to see a vanilla LSTM run
where lexical information is simply appended to the word vectors.

The authors end the paper with some helpful analysis of the
models. These experiments show that the model indeed learns
intensification and negation to some extent. In these experiments, it
would be interesting to know how the model behaves with
out-of-vocabulary words (with respect to the lexicons). Does the model
learn beyond memorization, and does generalization happen for words that
the model has not seen in training? Minor remark here: the figures and
tables are too small to be read in print.

The paper is mostly well-written apart from the points noted above.  It
could benefit from some proofreading as there are some grammatical
errors and typos left. In particular, the beginning of the abstract is
hard to read.

Overall, the paper pursues a reasonable line of research. The largest
potential issue I see is a somewhat shaky comparison to related
work. This could be fixed by including some stronger baselines in the
final model. For me, it would be crucial to establish whether
comparability is given in the experiments, and I hope that the authors
can shed some light on this in their response.

[YOG14] http://www.aclweb.org/anthology/P14-1074

--------------

Update after author response

Thank you for clarifying the concerns about the experimental setup. 

NSCL: I do now believe that the comparison is with Teng et al. is fair.

LSTM: Good to know that you did this. However, this is a crucial part of the
paper. As it stands, the baselines are weak. Marginal improvement is still too
vague, better would be an open comparison including a significance test.

OOV: I understand how the model is defined, but what is the effect on OOV
words? This would make for a much more interesting additional experiment than
the current regularization experiments.

[Official Review · Reviewer 2 · rating 4 · confidence 5]
soundness 5 · originality 3 · clarity 5 · impact 3 · substance 4 · appropriateness 5 · meaningful comparison 5 · presentation format Oral Presentation

- Strengths:
This paper proposes a nice way to combine the neural model (LSTM) with
linguistic knowledge (sentiment lexicon, negation and intensity). The method is
simple yet effective. It achieves the state-of-the-art performance on Movie
Review dataset and is competitive against the best models on SST dataset.    

- Weaknesses:
Similar idea has also been used in (Teng et al., 2016). Though this work is 
more elegant in the framework design and mathematical representation, the
experimental comparison with (Teng et al., 2016) is not as convincing as the
comparisons with the rest methods. The authors only reported the
re-implementation results on the sentence level experiment of SST and did not
report their own phrase-level results.

Some details are not well explained, see discussions below.

- General Discussion:

The reviewer has the following questions/suggestions about this work,

1. Since the SST dataset has phrase-level annotations, it is better to show the
statistics of the times that negation or intensity words actually take effect.
For example, how many times the word "nothing" appears and how many times it
changes the polarity of the context.

2. In section 4.5, the bi-LSTM is used for the regularizers. Is bi-LSTM used to
predict the sentiment label?

3. The authors claimed that "we only use the sentence-level annotation since
one of
our goals is to avoid expensive phrase-level annotation". However, the reviewer
still suggest to add the results. Please report them in the rebuttal phase if
possible.

4. "s_c is a parameter to be optimized but could also be set fixed with prior
knowledge."  The reviewer didn't find the specific definition of s_c in the
experiment section, is it learned or set fixed?  What is the learned or fixed
value?

5. In section 5.4 and 5.5, it is suggested to conduct an additional experiment
with part of the SST dataset where only phrases with negation/intensity words
are included. Report the results on this sub-dataset with and without the
corresponding regularizer can be more convincing.